# Metagenomic and Metabolomic Analyses Reveal the Role of a Bacteriocin-Producing Strain of *Enterococcus faecalis* DH9003 in Regulating Gut Microbiota in Mice

**DOI:** 10.3390/microorganisms13020372

**Published:** 2025-02-08

**Authors:** Zhiyu Fu, Haitao Zhang, Zhenzhu Yang, Yujun Liu, Peng Wang, Junjie Zhang, Hai Chi

**Affiliations:** 1Key Laboratory of Protection and Utilization of Aquatic Germplasm Resource, Liaoning Ocean and Fisheries Science Research Institute, Dalian 116023, China; hkyfzy@126.com (Z.F.); lyj7005@126.com (Y.L.); 2East China Sea Fisheries Research Institute, Chinese Academy of Fishery Sciences, Shanghai 200090, China; haitao2022220877@163.com (H.Z.); yzz9741@163.com (Z.Y.); wangp@ecsf.ac.cn (P.W.); 3School of Ocean Food and Biology Engineering, Jiangsu Ocean University, Lianyungang 222005, China; zjjdrl180@163.com

**Keywords:** *Enterococcus faecalis*, metagenomic sequence, metabolomic analysis, gut microbiota, probiotics

## Abstract

To investigate the regulatory effect of a bacteriocin-producing strain of *Enterococcus faecalis* DH9003 on the gut microbiota of mice, 15 healthy C57 male mice were randomly administered an equal volume of sterile normal saline (HD, control group, *n* = 7) and *E. faecalis* DH9003 (YD, treatment group, *n* = 8) via gavage. Metagenomic and metabolomic analyses were performed to determine the composition and metabolic function of the intestinal microbiota in mice. The results showed that the relative abundance of Firmicutes continuously increased over time in YD compared to HD. The number of *E. faecalis* DH9003 increased slowly and remained steady from days 7 to 28, indicating that *E. faecalis* DH9003 could colonize a considerable number of mouse guts via intragastric administration. Supplementation with *E. faecalis* DH9003 demonstrated a regulatory effect on the intestinal microbiota composition of mice, causing a shift in the relative abundance of Bacteroidetes and Firmicutes at the phylum level. In addition, a total of 2426 different metabolites were found in mouse feces, including 1286 and 1140 metabolites in positive and negative modes, respectively. Vitamin B6 and succinate were the most regulated and downregulated metabolites in negative ion mode, and the most upregulated and downregulated metabolites in positive ion mode were N-methyl-glutamic acid and N-octanoyl sphingosine. In conclusion, *E. faecalis* DH9003 can colonize mice gut, affecting the gut microbiota and metabolic competence. This strain therefore offers considerable potential for application as a probiotic.

## 1. Introduction

Lactic acid bacteria (LAB) are Gram-positive bacteria with a diverse morphological, metabolic, and physiological profile [1]. LAB have been considered as “Generally Recognized as Safe (GRAS)” because of their non-toxic, non-pathogenic, and non-carcinogenic properties [2]. LAB also have the capacity to create ribosome-synthesized antimicrobial peptides, known as bacteirocins, with excellent safety, good thermal stability, easy hydrolysis by proteases, and efficient inhibition of closely related bacteria [3,4]. Compared to traditional broad-spectrum antibiotics, which can disrupt the host microbiota and impair their function in a variety of ways, bacteriocins derived from LAB have specific antibacterial properties and efficiently kill target bacteria, while avoiding damage to the gut microbiota [1,5,6]. As a result, LAB are anticipated to have substantial implications for the treatment of intestinal disorders caused by some pathogenic bacteria.

The digestive tract is the main site of interaction between microorganisms and the human immune system, allowing naturally occurring bacteria to survive during opposing infections [7]. A full intestinal mucosal barrier can protect the organism from invasion by intestinal bacteria, endotoxins, and antigens, thus preserving its overall health [7]. Gut microbiota are responsible for the production of mucus layers. Under normal conditions, probiotics and pathogenic bacteria in the intestines are interdependent and mutually limited to achieve equilibrium and construct the microbiota barrier of the intestinal mucosa [8].

The highest proportion of obligate anaerobic bacteria in the intestine primarily protect intestinal health by adhering to the surface of the intestinal mucosa, inhibiting the adhesion of pathogenic bacteria through colonization antagonism and exerting antimicrobial and regulatory effects on the intestinal immune response [7,9,10]. By secreting secondary metabolites, such as lactic acid and short-chain fatty acids, the intestinal pH and redox potential are decreased, inhibiting the development of acid-resistant spoilage and pathogenic bacteria [11,12,13]. Bacteriocins from LAB secretion inhibit the colonization and proliferation of intestinal facultative anaerobic bacteria and foreign microorganisms, and the prevalent gut microbiota compete with negative microbes for nutrition, thereby inhibiting their growth [14].

LAB can effectively modulate the intestinal barrier function. First, LAB can stimulate epithelial cell proliferation and differentiation, thereby sustaining the regeneration and homeostasis of intestinal epithelial cells [14]. In a mouse model of enteritis, Wu et al. [15] discovered that *Lactobacillus reuteri* activates the Wnt/β-catenin pathway by boosting the production of R-spondins and promoting the proliferation of intestinal epithelial cells. Second, LAB can lower the intestinal barrier’s permeability. Under normal conditions, the permeability of the intestinal barrier is selective, allowing vital nutrients and water to pass, while preventing the passage of poisons and pathogens. Yi et al. [16] discovered that feeding *Lactobacillus reuteri* to weaned pigs can successfully maintain the integrity of the mucosal membrane by upregulating the production of TJ proteins in host intestinal epithelial cells. Thirdly, LAB can increase the expression and secretion of mucin genes. Rokana et al. [17] investigated the functional efficiency of *Lactobacillus plantarum* in Salmonella-infected mice and discovered that ingesting fermented milk derived from *Lactobacillus plantarum* MTCC 5690 boosted MUC2 expression and improved intestinal barrier function. Fourth, LAB can establish mucosal colonization sites. LAB bind to intestinal mucus via cell surface proteins and pili, inhibiting and excluding bacterial enteric pathogens, thereby maintaining intestinal homeostasis and supporting intestinal health [18,19]. Kim [20] identified lectins from the cell surface proteins of *Lac. brevis* FSB-1, which can detect and bind to the terminal sugar chains of rat colon mucus, limiting pathogenic bacteria adherence. Finally, LAB can regulate the composition of intestinal microbiota [21,22]. The human gastrointestinal tract has a complex and dynamic microbial ecology consisting of trillions of bacteria. According to previous studies, Gut microbiota and their metabolites have been shown to influence the formation, homeostasis, and function of innate and adaptive immune cells; prevent the growth of dangerous bacteria; and maintain intestinal epithelial cell integrity [23,24,25].

*Enterococcus* sp. is a type of LAB with high acid resistance and colonization ability, low nutritional needs, ease of culture, and widespread dispersion. It is mostly present in the gastrointestinal tract, oral cavity, food, and other animal tissues, and may be used as a probiotic to regulate the intestinal microenvironment and immune system [26,27]. In our previous study, *E. faecalis* DH9003 was found to possess bacteriocin characteristics and was identified as a potential probiotic without harmful effects in mice [28]. Therefore, further research on the role of *E. faecalis* DH9003 in the regulation of gut microbiota is needed. The aim of this study was to investigate the regulatory effect of a bacteriocin-producing strain of *E. faecalis* DH9003 on the gut microbiota of mice. We used C57 male mice as a model and introduced *E. faecalis* DH9003 into the mice by oral gavage. Subsequently, the microbial diversity in mice feces was analyzed using high-throughput sequencing technology, and we employed non-targeted metabolomics technology based on UHPLC-Q-TOF-MS to study changes in secondary metabolites in mice feces after different treatments over time. The results provide insights into the regulatory roles of *E. faecalis* DH9003 in the gut microbiota of mice, as well as the further probiotic development of bacteriocin-producing strains as a probiotic.

## 2. Materials and Methods

### 2.1. Materials and Strain Culture Conditions

*E. faecalis* DH9003, a bacteriocin-producing strain, and *Listeria monocytogenes* LFM2813, an indicator strain, were incubated in Brain Heart Infusion (BHI, Oxoid, Hampshire, England) at 30 °C for 24 h.

Four-week-old male C57 mice, weighing of 15 ± 1 g, were purchased from the Experimental Animal Management Department of the Shanghai Family Planning Research Institute (Shanghai, China). Mixed feed for mice was purchased from Shanghai Xinhui Animal Feed Co., Ltd. (Shanghai, China).

The gavage needle (size 8, 45 mm), sodium chloride, formaldehyde, 100% ethanol, xylene, neutral gum, sterile normal saline, and phosphate-buffer solution (PBS, pH 7.2) were purchased from Sinopharm Chemical Reagent Co., Ltd. (Shanghai, China).

### 2.2. E. faecalis DH9003 Preparation

Approximately 2% of the overnight culture of *E. faecalis* DH9003 was transferred to BHI broth and incubated at 30 °C for 12 h. The precipitation from the culture was collected by centrifugation at 2000× *g* for 5 min and washed 3 times with sterile normal saline. The precipitation was suspended in sterile normal saline to bring the colony number of *E. faecalis* DH9003 to 10^9^ CFU/mL. The fresh culture was maintained at 4 °C for gastric perfusion.

### 2.3. Fecal Sample Collection

A total of 15 male C57 mice were acclimated in M1 cages (290 × 180 × 160 mm) and fed for 7 days in an animal room maintained at a temperature of (23 ± 2) °C, with a relative humidity ranging from 50% to 60%, following approval from the Ethics Committee of Liaoning Ocean and Fisheries Science Research Institute (Approval code: 2022-12-01EC-008). The mice were subjected to a 12 h light/dark cycle (8 am to 8 pm, exposed to a 15-watt incandescent light source), allowing them to eat and drink freely.

After a week of acclimation, the mice were randomly assigned to either a control group (marked as HD group, *n* = 7) or an *E. faecalis* DH9003 treatment group (marked as YD group, *n* = 8). Each group of mice was fed with pre-disinfected SPF feed and sterilized water daily. In addition, the YD group was given 0.2 mL/10 g of body weight of *E. faecalis* DH9003 by gavage daily, whereas the HD group was given a corresponding dose of sterile normal saline via gavage once daily. The experiment lasted 4 weeks, during which the mice feces were collected regularly and stored at −80 °C.

### 2.4. Detection of Antimicrobial Activity

About 0.1 g of fresh mouse feces, under sterile conditions, was added to a centrifuge tube containing 0.9 mL sterile normal saline. The mixture was shaken with a vortex and then properly 10-fold diluted from 10^−7^ to 10^−9^. The samples from each gradient were spread on BHI soft agar (BHI broth containing 0.6% agar powder, BHISA) and incubated at 30 °C for 24 h. After a 24 h incubation, 100 μL of an overnight culture of the indicator strain (*Listeria monocytogenes* LMF2813) was transferred to 5 mL of BHISA. The mixture was then spread on the same plate and incubated at 30 °C for 24 h. Plate(s) with clear antimicrobial inhibition zone(s) after incubation indicated the existence of antimicrobial activity. Colonie(s) in the clear zone(s) were collected and sent for PCR identification according to the procedure described by Chi et al. [3].

### 2.5. DNA Isolation

The DNA isolation from mouse feces in the YD and HD groups was conduct according to the instruction provided by the GenElute Bacterial Genomic DNA kit (Sigma-Aldrich, St. Louis, MO, USA). The purity and concentration of DNA were detected using 1.2% agarose gel electrophoresis. The genomic DHA was used as template for PCR amplification. The primers and PCR program were conducted according to the method described by Yang et al. [28]. The PCR products were sent to Sangong Biotech (Shanghai, China) for bacterial identification.

### 2.6. Metagenomic Sequencing Analysis

The DNA extracted from mouse feces was amplified using primers from the V3-V4 regions according to the method previously described by Kong et al. [29]. The accuracy of PCR products was detected using 2% agarose gel electrophoresis, and the PCR products were recovered and purified. The Quant-iT PicoGreen dsDNA detection kit (Invitroge™, Waltham, MA, USA) was used for the quantitative analysis, the DNA fragment with an average size of around 0.4 kbp was used for the library construction. A paired-end library was constructed by using Miseq library. Adapters with a full complement of sequencing primer hybridization sites were ligated to the blunt ends of fragments. Sequencing was performed on an Illumina NovaseqPE250 platform (Illumina Inc., San Diego, CA, USA) for the microbial diversity analysis.

### 2.7. Sequencing Quality Control

The original offline data of high-throughput sequencing were checked for sequence quality and the issue samples were retested. The original sequence that passed the preliminary quality check was then divided into a library and sample based on the tag sequence information, and the tag sequence was removed accordingly. Based on the QIIME 2 analysis procedure, sequence de-primers, quality filtering, and de-noising were performed to obtain effective sequences, which were then spliced and chimera filtered to yield high-quality sequences for further analysis [30]. Briefly, the adaptors were trimmed from paired-end reads, and low-quality reads (length < 50 bp or with a quality value < 20 or having N bases) were removed. Contigs with a length of at least 0.3 kbp were collected as the final assembled results. Those contigs were then used for gene prediction and annotation.

### 2.8. Metabolomic Analysis

The metabolomic analysis of mouse fecal samples was performed using ultra-performance liquid chromatography tandem time-of-flight mass spectrometry (UHPLC-Q-TOF-MS) [31,32]. Briefly, after progressively thawing at 4 °C, approximately 0.5 g of sample was added to a precooled methanol/acetonitrile/water solution (2:2:1, *v*/*v*) and thoroughly mixed. After 30 min of low-temperature ultrasound, the sample was held at −20 °C for an additional 10 min. The supernatant from the sample was vacuum dried after centrifuging at 14,000× *g* for 20 min at 4 °C. The supernatant was mixed with 100 μL of a cetonitrile aqueous solution (acetonitrile: water = 1:1) and subsequently centrifuged at 14,000× *g* at 4 °C for 15 min. The supernatant was used for the metabolomic analysis.

The mass spectrometry conditions in this study are as follows: An AB Triple TOF 6600 mass spectrometer (Applied Biosystems Inc, Foster City, CA, USA) was used to collect the primary and secondary spectra of the samples. An electrospray ionization (ESI) source was used to collect data in positive and negative ion detection modes. The scanning range was 60–1000 Da, bombardment energy was 30 ± 14 EV, ion source temperature was set to 600 °C, and ionization source voltage was 5500 V.

### 2.9. Statistics Analysis

QIIME 2 was used to match the sequences with the Greenenes and Silva databases and to annotate the sequence categorization information. The taxonomic composition was evaluated using QIIME 2. The non-metric multidimensional scaling analysis (NMDS) was conducted on the Bray–Curtis distance matrix in R, and the compositional differences across microbial communities were shown using a two-dimensional ordination diagram. The quantitative analysis of biomarkers in different groups was performed using the Python LEfSe package and R (https://www.bioconductor.org/packages/release/bioc/html/lefser.html, accessed on 23 January 2025).

The supervised orthogonal partial least squares discriminant analysis (OPLS-DA) model was used to investigate the differential lipid substances associated with grouping from the data set. The variable weight for the project (VIP) > 1 and *p* < 0.05 were used as the screening criteria for differential metabolites. Metabolic pathways were annotated using the KEGG database (http://www.genome.ad.jp/kegg/, accessed on 23 January 2025) to determine the metabolic processes in which differential metabolites were involved.

## 3. Results

### 3.1. Metagenomic Sequencing Data

In this study, fecal samples from different treatment groups were collected at various time points (0, 3, 7, 14, 21, and 28 d). Each group was further divided into three subgroups and 36 fecal samples were collected from the mice. Qualified sequence data, following the quality control of the metagenomic sequencing results, were distributed between 79,193 bp and 93,801 bp. Because of strict quality control and paired-end read trimming, high-quality sequence data were obtained ranging from 43,303 bp to 56,408 bp; the average length of high-quality sequences (length < 50 bp or with a quality value < 20 were discarded) was distributed in 424 to 430 bp, and the operational taxonomic unit (OTU) of each sample was distributed between 6168 and 7295 (Appendix A).

### 3.2. Alpha Diversity Analysis

Alpha diversity reflects species variety in a particular ecosystem or ecological environment [33]. Sparse curves are commonly used in the microbiome to assess the saturation of the sequencing volume or sample size, specifically to determine whether the sequencing volume is sufficient. When the sequencing depth of random sampling steadily increased and the curve no longer increased considerably, the sequencing volume was regarded as sufficient [31]. If the sequencing volume increases, the alpha diversity index of the sample will not vary considerably. At this point, the sample’s alpha diversity index remained steady. It can be used to assess the species richness of several sequencing samples and to determine whether the sample size is appropriate. Figure 1 shows sparse sequencing curves for HD and YD. Figure 1 showed that the slope of the curve was smooth, and the change was minor when the sample’s sequencing depth increased to a certain point, indicating that the sample size was appropriately adjusted. Simultaneously, as the sequencing depth increased, the species richness of YD decreased relative to that of HD. This might be due to the continuous entry of high-dose *E. faecalis* DH9003 into the intestines in a short period of time, where it colonizes and becomes a relatively dominant microbiota, inhibiting other bacteria and resulting in reduced species richness. This is consistent with the results reported by Yang et al. [34], who found that the addition of compound probiotics resulted in lower species richness than the control group.

### 3.3. Beta Diversity Analysis

Beta diversity refers to the diversity of species composition or the rate of species replacement along the environmental gradient across different groups that alter along the environmental gradient; it is also known as inter-habitat diversity [35]. Non-constrained sorting methods such as principal coordinate analysis (PCoA) and non-metric multidimensional scaling (NMDS) analysis may minimize the dimension of the microbiological data, and the distribution of samples on the continuous sorting axis can show a general pattern of data change [36]. The clustering technique may identify the selection of discontinuous items in the environment while classifying the data. This study used NMDS analysis to identify variations between samples [36]. The NMDS analysis can directly determine whether there is an obvious difference between biological groupings and similarities within groups. As shown in Figure 2, each point represents a sample, with various colors representing different groups. The differences in the microbial community between the two samples decreased as the distance between them decreased. In contrast, there was greater variation in microbial community composition between the samples. In this study, the sample distance between the blank group without gavage and the *E. faecalis* DH9003 treatment group was low, but the sample distance after gavage was broad, indicating that the experiment was successful. At the same time, as the gavage days varied, the sample spread further, demonstrating that *E. faecalis* DH9003 gavage has an effect on the spread of intestinal microbiota in mice, which requires additional study in the future.

### 3.4. Microorganism Composition Analysis

Figure 3 depicts the relative richness of the microbial communities at the phylum level in mouse feces from various treatment groups at different time points. The results revealed that Bacteroidetes and Firmicutes were the dominant phyla in mouse feces, with a total relative abundance of more than 93%, indicating that exogenous supplemental probiotic consumption has a regulatory effect on the intestinal microbiota structure of mice but cannot change the main dominant microbiota, which is consistent with the research results of Pan et al. [37]. At the same time, the difference at the phylum level of the YD group appears primarily in the alterations of Bacteroidetes and Firmicutes, which is consistent with the findings of Vemuri et al. [38]. The relative abundances of Firmicutes in the HD group were 37.13% on day 0, 51.67% on day 7, and 41.86% on day 28. The relative abundances of Firmicutes in the YD group were 40.17% on day 0, 47.43% on day 7, and 63.58% on day 28. In the HD group, the relative abundance of Firmicutes first increased and then decreased with an increase in feeding time, indicating that the new environment may stimulate a change in the relative abundance of Firmicutes in the intestinal microbiota. However, after a period of adaptation, the relative abundance of Firmicutes in the intestinal microbiota decreased and returned to stable levels. The relative abundance of Firmicutes increased in the YD group as *E. faecalis* was introduced intragastrically, indicating that *E. faecalis* DH9003 (Firmicutes) may colonize the intestinal tract of mice after ingestion and may gradually become the dominant intestinal microbiota.

### 3.5. LDA Effect Size (Lefse) Analysis and Marker Species Determination

The LDA effect size (Lefse) analysis is a difference analysis approach that can immediately assess the differences in all categorization levels at the same time and identify the most intense differential species across groups, also known as biomarkers. There were no limits to evaluating the variations in the community composition between the sample groups [38]. Lefse may go deep into distinct subgroups and identify signal microbial communities that are constant across subgroups. This analytical approach is currently widely employed in the disciplines of microbial amplicon analysis, metagenomic analysis, etc., and is particularly suitable for identifying biomarkers in microbiological research. Figure 4 depicts the taxonomic cladistic diagram of the mice treated with *E. faecalis* DH9003, showing the taxonomic hierarchy of the sample community from phylum to genus (inner circle to outer circle), where the node size corresponds to the average relative abundance of the taxon, the hollow node represents the taxon with no significant component difference, and the blue node indicates that these taxa reflect significant component differences and are highly abundant. These results further demonstrated that *Enterococcus* sp. was the dominant species in the treatment groups. Thus, the supplementation of *E. faecalis* DH9003 has the potential to affect both the variety and number of intestinal microbiota in mice.

### 3.6. E. faecalis DH9003 Isolation and Its Antimicrobial Activity

After 28 days of treatment with *E. faecalis* DH9003, the strains from the fecal samples were spread on agar plates. The antimicrobial activity assay revealed clear antibacterial zones at the 10^7^ and 10^8^ dilutions (Appendix A), but the fecal samples from the control group exhibited no antimicrobial activity. The strains that showed antibacterial activity against *Listeria monocytogenes* on the plates were selected and sequenced. The sequencing results revealed that the strains were *E. faecalis*, with 99.9% similarity. These findings suggest that *E. faecalis* DH9003 colonizes mouse guts and likely performs a regulatory function in pathogen inhibition.

### 3.7. Metabolic Analysis

PCA analysis was performed on the peaks extracted from all experimental and QC samples, as shown in Appendix A. The experimental results indicated that the QC samples were tightly clustered together in both positive and negative ion modes, indicating that the experiment was highly reproducible.

In this study, 2426 metabolites were identified, including 1286 and 1140 metabolites from the positive and negative modes, respectively (Appendix A). All the identified metabolites were classified according to their chemical taxonomy, and the proportions of all metabolites are shown in Figure 5. The results illustrated that lipids and lipid-like molecules, as well as organic acids and their derivatives, were the top two metabolites, accounting for more than half of all metabolites (27.61% and 23.94%, respectively). Organheterocyclic compounds, benzenoids, and organic oxygen compounds accounted for 10.47, 8.24, and 7.21% of the total metabolites, respectively. Other metabolites accounted for less than 5% of the total. There were approximately 13.76% unidentified metabolites, which accounted for approximately 13.76% of the total.

To discriminate between YD_0 and YD_28, the OPLA-DA method was first applied to both groups. Appendix A depicts the OPLA-DA score chart and replacement test chart for YD_0 against YD_28. The samples from each group may be well-clustered into one group, with a clear differentiation between the two groups, indicating that the experiment has strong repeatability. At the same time, in the positive ion mode, R2X = 0.601, R2Y = 1, and Q2 = 0.99, and in the negative ion mode, R2X = 0.761, R2Y = 0.999, and Q2 = 0.996, demonstrating that the model is stable and reliable. Replacement and retrograde tests were performed to ensure the model’s efficacy. To avoid fitting a supervised model during the modeling phase, tests were conducted to ensure the model’s efficacy. Figure 6 also shows that R2 and Q2 of the random model steadily declined as replacement retention decreased, demonstrating that the original model failed to show overfitting and was robust.

Univariate statistical analysis investigates intra-group variability and inter-group differences at the level of a single variable, whereas multidimensional statistical analysis addresses inter-group differences and intra-group variability at the aggregate level. Figure 7 depicts the volcano pattern diagram of the univariate statistical analysis based on the positive and negative ion mode detection. The results showed that the metabolites were more elevated in the negative ion mode than in the positive ion mode in comparison with YD_28 and YD_0. On comparing the metabolites of YD_28 to HD_28, the opposite findings were obtained; the metabolites were less elevated in the negative mode and more elevated in the positive mode.

Metabolome data contain a high level of complexity and correlation across factors. Traditional univariate analysis is incapable of rapidly and properly identifying possible information within data. Multivariate statistical approaches, such as PCA and OPLS-DA analysis, are required to decrease the dimensionality of the gathered multidimensional data while retaining as much original information as feasible. Therefore, we also evaluated the differential metabolites of YD_0 and YD_28 using OPLS-DA (variable importance in projection > 1 and *p* < 0.05) to identify significant differences. The top 10 upregulated and downregulated metabolites of YD_0 and YD_28 in mouse fecal samples after treatment with *E. faecalis* DH9003 using negative and positive ion modes are shown in Figure 8. Vitamin B6 was the most upregulated metabolite in negative ion mode, with a more than 1000-fold change. The most downregulated metabolites in negative ion mode were succinate, urushiol II, and polygalic acid. In contrast, N-methyl-glutamic acid and N-octanoyl sphingosine were the most upregulated and downregulated metabolites, respectively, in positive ion mode.

The KEGG analysis was based on the KEGG pathway as a unit, with the metabolic pathways involved by the species or closely related species as the background. Fisher’s exact test was used to analyze and calculate the significant level of metabolite enrichment in each pathway, thereby determining the metabolism and signal transduction pathways that are significantly affected [39]. Figure 9 depicts the KEGG enrichment pathways and the top 20 differentially expressed metabolites with the highest significance. The vertical axis represents each KEGG metabolic pathway, the horizontal axis represents the number of differentially expressed metabolites contained in each KEGG metabolic pathway species, and the color represents the *p*-value of the enrichment analysis. The deeper the coloring, the lower the *p*-value and the higher the degree of enrichment. The results showed that the metabolic pathways with the highest significant levels in the figure were the ABC transporter, biosynthesis of amino acid, central carbon metabolism, purine metabolism, and protein digestion and absorption.

## 4. Discussion

The gut microbiota are microorganisms that live in the digestive tract, including the esophagus, stomach, and intestines, and play a vital role in gut health [7]. Most studies on the gut microbiota have been conducted in the digestive organs, specifically the stomach, small intestine, and large intestine. The oral administration of bacteriocins via LAB producers, which can withstand acidic conditions in the stomach, allows us to avoid proteolysis during gastric transit and to create bacteriocins in situ in the large intestine [40]. Therefore, we conducted our study using fecal samples from mice.

*Enterococcus* sp., a well-known probiotic in LAB, has been shown to improve the health and management of bacterial infections. In this study, we used *E. faecalis* DH9003, with its bacteriocin-like and probiotic characteristics, to determine its influence on gut microbiota composition and metabolism in mice. We found that *E. faecalis* DH9003 supplementation significantly altered the gut microbiota and increased microbial diversity and richness during treatment. These findings demonstrated that Bacteroidetes and Firmicutes were the dominant phyla, with the number of Firmicutes progressively increasing during the course of the experiment. *E. faecalis* DH9003 was also identified in fecal samples at a dilution of 10^8^, indicating that *E. faecalis* DH9003 shows the potential to significantly affect the gut microbiota composition. Pan et al. [37] orally induced LAB in infected mice to reduce symptoms, demonstrating that LAB may effectively modify gut microbiota composition, activate immune systems, and thereby alleviate the symptoms. Several studies have proposed that probiotics derived from LAB can adhere to intestinal epithelial cells, preventing pathogen adhesion and producing antimicrobial peptides, such as bacterocins [41,42]. The processes by which probiotics work and how they affect the immune system responses are highly complicated. Future research should focus on determining the associations among probiotic intake, specific bacterial taxa, and the immune system.

Metabolomics is the study of the species, quantities, and change behaviors of metabolites (endogenous metabolites) with molecular weights less than 1.5 kDa generated by external stimuli, pathophysiological alterations, and gene mutations [43,44]. Metabolomics technology was used to analyze the differences in metabolic levels between the experimental and control groups; to identify different metabolites, which is beneficial in biomarker screening; to study the biological processes involved in the various metabolites; to reveal the mechanism of life activities involved; and to complete regulatory pathway research [32]. In this study, ultra-performance liquid chromatography tandem time-of-flight mass spectrometry (UHPLC-Q-TOF MS) was used to analyze fecal samples from mice. The structure of the metabolites in the samples was identified by matching the retention time, molecular weight, secondary fragmentation spectrum, collision energy, and other information of the metabolites in the database. Metabolites with significant changes in mice after treatment with *E. faecalis* DH9003 were discovered, allowing us to further investigate the changes in metabolites in mice after treatment with *E. faecalis* DH9003 and their related targets and pathways.

Vitamin B6 is found in the body in several forms, including pyridoxine (PN), pyridoxal (PL), pyridoxamine (PM), pyridoxine phosphate (PLP), pyridoxal phosphate (PNP), and pyridoxamine phosphate (PMP), and it plays an important role in metabolic processes such as amino acid metabolism and neurotransmitter production. Vitamin B6 is necessary for the healthy development and appropriate physiological activities in the human body [45,46]. It acts as a coenzyme factor and participates in over 140 biochemical reactions in cells, including key metabolic reactions such as amino acid synthesis, interconversion, degradation, carbohydrate, lipid, nucleic acid metabolism, and heme synthesis [47]. In our study, vitamin B6 was the most upregulated metabolite, indicating that vitamin B6 was likely involved in multiple metabolic processes in mice. Ouyang et al. [45] induced vitamin B6 into broilers as dietary supplementation, identifying 83 distinct metabolites associated with vitamin B6 amelioration, which were primarily enriched in glycerophospholipid, caffeine, and glutathione metabolism. Lipid and lipid-like molecules and nucleosides, nucleotides, and analogs were elevated in both the negative and positive ion modes. The beneficial functions of vitamin B6 metabolism may lead to improved metabolic growth and development in mice. Furthermore, esters, such as succinate, α-tochopheryl acetate, and α-tocopheryl acetate, were dramatically downregulated in mice. The reduction in esters demonstrated that supplementation with *E. faecalis* DH9003 had obvious health benefits. Further investigation into the causal relationship between *E. faecalis* DH9003 supplements and health is likewise a priority.

## 5. Conclusions

The present study demonstrated that *E. faecalis* DH9003 supplementation in mice could considerably modify the variety and functional enrichment of metabolic pathways in the gut microbiota after 7 days of feeding. Firmicutes, particularly substituted Bacteroidetes, gradually became the dominant phylum after 7 days of supplementation with *E. faecalis* DH9003. Moreover, 2426 different metabolites, including 1286 and 1140 metabolites in positive and negative modes, were found after 28 days. The KEGG analysis revealed the pathways underlying the ABC transporter, biosynthesis of amino acid, central carbon metabolism, purine metabolism, and protein digestion and absorption. These findings might help to better understand the regulatory functions of *E. faecalis* DH9003 in gut microbiota modification and metabolic alterations in mice, as well as its prospective applications as a probiotic in the food industry.

## Figures and Tables

**Figure 1 microorganisms-13-00372-f001:**
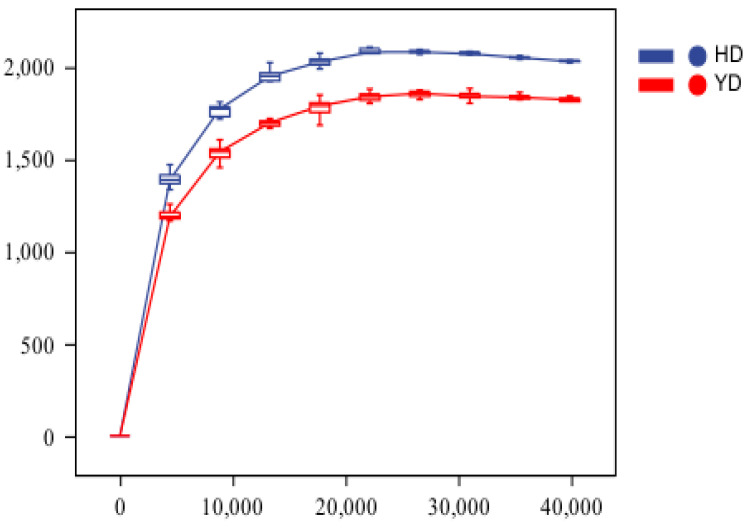
Sparse curves of samples in different treatment groups. HD serves as the control group, and YD serves as the experimental group that was treated with E. *faecalis*.

**Figure 2 microorganisms-13-00372-f002:**
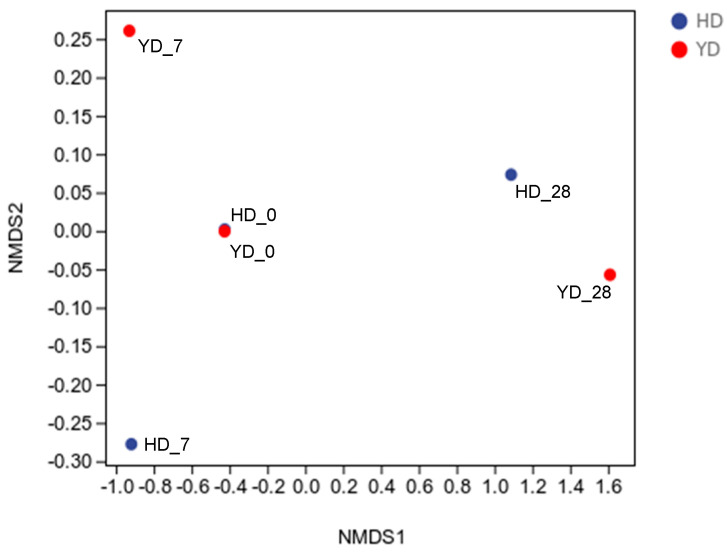
Beta diversity analysis. HD serves as the control group; HD_0, HD_7, and HD_28 represent the samples collected at 0, 7, and 28 days. YD serves as the experimental group that was treated with E. *faecalis* DH9003. YD_0, YD_7, and YD_28 represent the samples collected at 0, 7, and 28 days, respectively.

**Figure 3 microorganisms-13-00372-f003:**
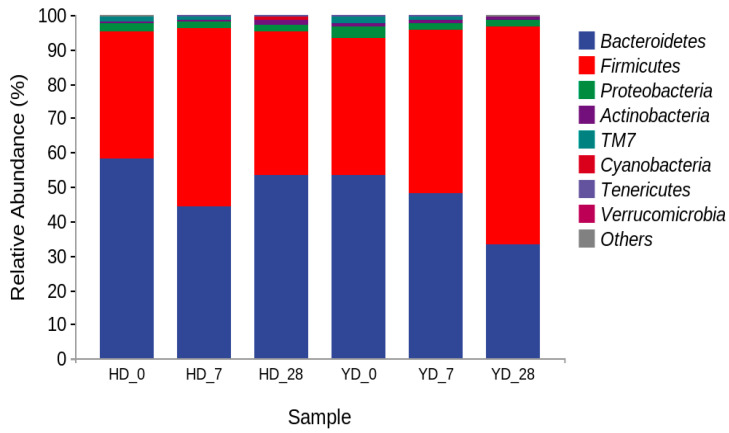
Relative abundance of microbial microbiota in feces of mice at different times in different treatment groups (HD and YD) at the phyla level. HD serves as the control group; HD_0, HD_7, and HD_28 represent the samples collected at 0, 7, and 28 days. YD serves as the experimental group that was treated with *E. faecalis*. YD_0, YD_7, and YD_28 represent the samples collected at 0, 7, and 28 days, respectively.

**Figure 4 microorganisms-13-00372-f004:**
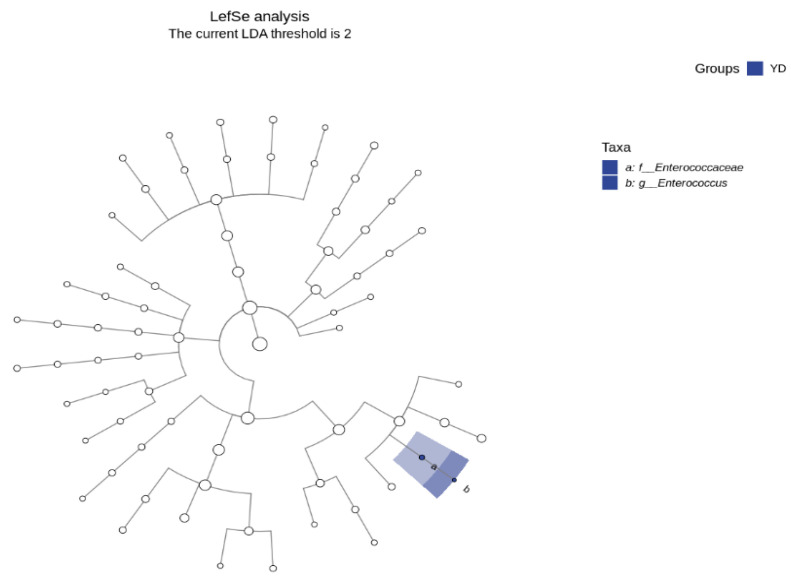
Taxonomic cladistic diagram of the mice treated with *E. faecalis* DH9003.

**Figure 5 microorganisms-13-00372-f005:**
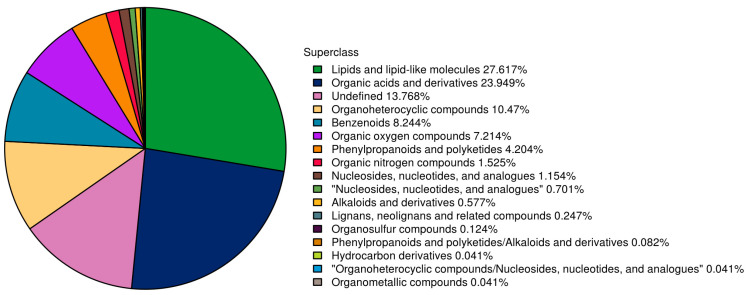
Proportion of identified metabolites in each chemical taxonomy from fecal samples.

**Figure 6 microorganisms-13-00372-f006:**
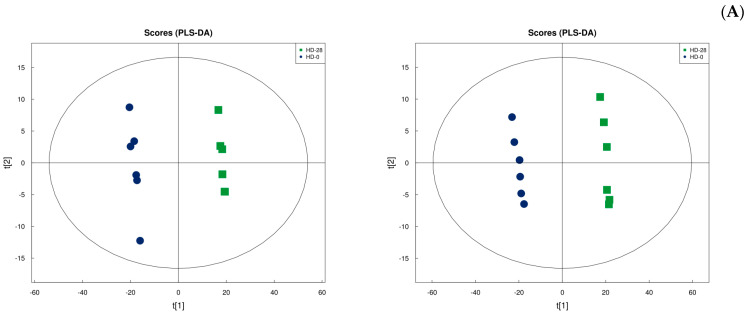
OPLS-DA score and permutation test chart of YD_0 against YD_28. YD serves as the experimental group that was treated with *E. faecalis* DH9003. YD_0 and YD_28 represent the samples collected at 0 and 28 days, respectively. (**A**) Positive ion mode, (**B**) negative ion mode.

**Figure 7 microorganisms-13-00372-f007:**
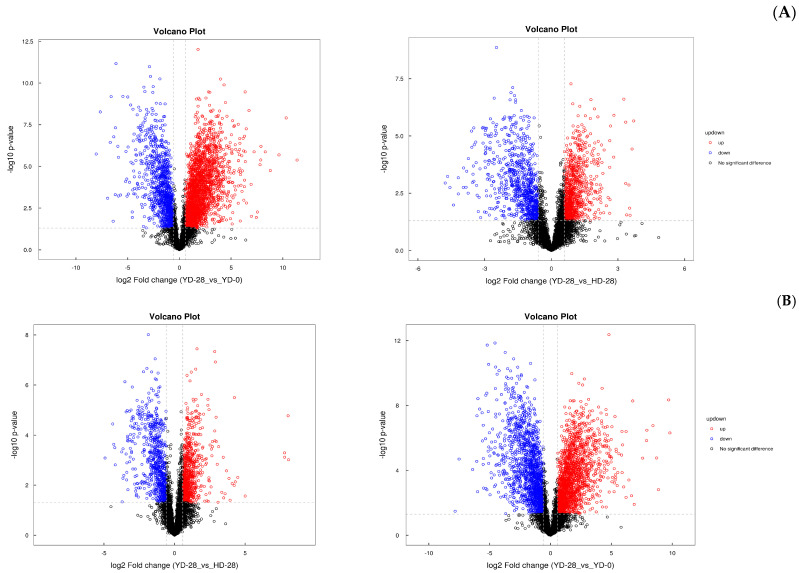
Volcanic pattern diagram. The horizontal axis in the figure represents the logarithm of log2 for fold change, and the vertical axis represents the logarithm of log10 for the significant *p*-value. Metabolites with significant differences: metabolites with FC > 1.5 and *p*-value < 0.05 are represented in red, while metabolites with FC < 0.67 and *p*-value < 0.05 are represented in blue. Non significantly different metabolites are represented in black. (**A**) Negative ion mode, (**B**) positive ion mode.

**Figure 8 microorganisms-13-00372-f008:**
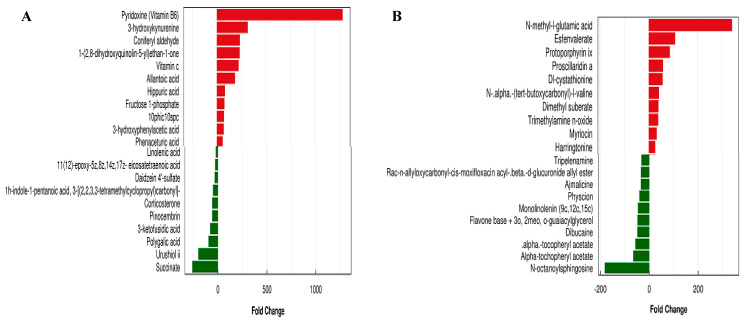
Significant differential metabolite expression fold change analysis of YD_28 and YD_0. (**A**) Negative ion mode, (**B**) positive ion mode.

**Figure 9 microorganisms-13-00372-f009:**
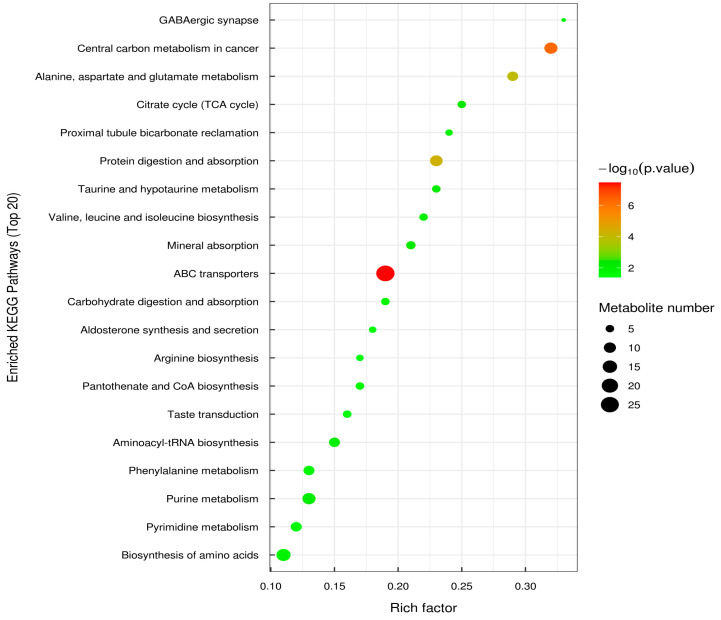
KEGG-enriched pathway bubble plot (top 20) of differentially expressed metabolites of YD_28 and YD_0.

## Data Availability

The original contributions presented in the study are included in the article, further inquiries can be directed to the corresponding author.

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
