# Peer review of "Metagenomic and Metabolomic Analyses Reveal the Role of a Bacteriocin-Producing Strain of Enterococcus faecalis DH9003 in Regulating Gut Microbiota in Mice"

_microorganisms, 2025, doi:10.3390/microorganisms13020372_

Round 1
Reviewer 1 Report (Previous Reviewer 1)
Comments and Suggestions for Authors
Can be accepted
Reviewer 2 Report (Previous Reviewer 3)
Comments and Suggestions for Authors
The authors have incorporated amendments in accordance with the comments previously submitted.
This manuscript is a resubmission of an earlier submission. The following is a list of the peer review reports and author responses from that submission.
Round 1
Reviewer 1 Report
Comments and Suggestions for Authors
The study uses a small sample size (15 mice), raising concerns about the conclusions' statistical power and robustness.
The metabolomic data identified 254 differential metabolites; however, the thresholds (VIP > 1, p < 0.05) used for selection should be justified further. Additionally, pathway enrichment analysis should include more details about how metabolite alterations align with functional outcomes.
Figures 1-6 are informative but require clearer legends. For example, Figure 6 should specify the top KEGG pathways and metabolites driving the enrichment analysis. Additionally, potential weight-loss benefits of E. faecalis DH9003, are speculative.
Author Response
Dear reviewer,
The comments provided by you were very helpful to us. Based on your comments, we have carefully revised the manuscript (point by point). The revised parts are marked in the manuscript, and responses to the reviewers' comments are below.
1.The study uses a small sample size (15 mice), raising concerns about the conclusions' statistical power and robustness.
A:Thanks for your comments.
The minimum sample size required for metabolomics and metagenomic analysis has no fixed standard value and could be impacted by a variety of criteria such as research objectives, sample heterogeneity, technological platforms and statistical methods utilized, and the expected effect size of the study. To obtain statistically significant findings, at least three technical duplicates of each sample should be performed to increase the results' reliability and repeatability. Furthermore, the number of biological replicates is determined by the projected effect magnitude and variability, with each treatment group generally requiring at least 3-6 biological replicates.
Specifically for the case of 15 mice in our study, we included 7-8 samples in each group and proper technical and conducted biological duplicates, these results may be sufficient. importantly, we re-divided the fecal samples into 3 subgroups, therefore, there were 24x6 samples total in the study.
2.The metabolomic data identified 254 differential metabolites; however, the thresholds (VIP > 1, p < 0.05) used for selection should be justified further. Additionally, pathway enrichment analysis should include more details about how metabolite alterations align with functional outcomes.
A: Thanks for your comments, we have added more information and figures, results related to your concerns in the manuscript.
3. Figures 1-6 are informative but require clearer legends. For example, Figure 6 should specify the top KEGG pathways and metabolites driving the enrichment analysis. Additionally, potential weight-loss benefits of E. faecalis DH9003, are speculative.
A:Thanks for your comments, we have changed the figures to make them clearer and added more figures and information. Additionally, potential weight-loss benefits of E. faecalis DH9003, are speculative. you are correct, thats why we changed the sentence to obtiain "healthy benefit" in line 486.
Reviewer 2 Report
Comments and Suggestions for Authors
The study explores the probiotic potential of E. faecalis DH9003 using metagenomic and metabolomic analyses, offering interesting insights into gut microbiota modulation. However, key improvements are needed. Why was a sample size of 15 mice considered sufficient, and how might it affect the statistical power? The reliance on 16S rRNA sequencing limits taxonomic resolutioncould whole-genome sequencing provide more comprehensive insights? The reduced species richness and dominance shift toward Firmicutes are noted but underexplored, what mechanisms might explain these changes? Furthermore, the clinical relevance of N-methyl-L-glutamate upregulation and the potential safety risks of long-term E. faecalis colonization need further discussion. By addressing these points and implementing larger sample sizes, robust statistical validation, and a stronger contextualization of the findings, this article can achieve greater clarity and impact.
Here my suggestions for specific improvements of the manuscript:
• Abstract: Include quantified results for microbial shifts and metabolite changes. • Line 15: Specify randomization and blinding methods. • Line 27: The term "micorbiota" should be corrected to "microbiota" in the keywords and elsewhere in the text. • Line 97: Justify the limitations of 16S rRNA sequencing and suggest alternatives. • Line 214: Provide hypotheses or explanations for the reduced species richness. • Line 297: Discuss criteria for metabolite selection and the implications of excluding unidentified ones. • Line 394: Address safety concerns related to the colonization of E. faecalis. • Line 403: Compare findings with similar studies for better contextualization. • Table 1: Modify the title in "Differential Metabolites Identified in Fecal Samples of Mice Treated with Enterococcus faecalis DH9003." • Figures 1 and 3, which show microbial diversity and phylum-level composition, could be presented together to allow direct comparisons between alpha diversity and relative abundances of key phyla. Include more detailed figure legends to explain the significance of each result, which would help readers interpret the findings more efficiently.Through these modifications, the article can be significantly improved, offering stronger support for its conclusions and enhancing its contribution to the field.
Comments on the Quality of English Language
minor editing is required
Author Response
Dear reviewer,
Your comments were very helpful to us. Based on your comments, we have carefully revised the manuscript (point by point). Firstly, we changed the abstract according to your suggestion, and also discussed more about the metabolites in the discussion part. The revised parts are marked in the manuscript, and responses to the reviewers' comments are below.
1. Abstract: Include quantified results for microbial shifts and metabolite changes.
A: Thanks for your comments, we have added more information in line 18-22, 24-29.
2. Line 15: Specify randomization and blinding methods.
A: Thanks for your comments, we added a "randomly" in line 15.
3. Line 27: The term "micorbiota" should be corrected to "microbiota" in the keywords and elsewhere in the text.
A:Thanks for your comments, we have carefully checked out the whole manuscript, and corrected the word to "microbiota".
4. Line 97: Justify the limitations of 16S rRNA sequencing and suggest alternatives
A: Thanks for your comments, we have added more information about the concerns in line 364-372.
5. Line 214: Provide hypotheses or explanations for the reduced species richness.
A:Thanks for your comments, we have explained the reduction of species richness in line 212-220.
6. Line 297: Discuss criteria for metabolite selection and the implications of excluding unidentified ones.
A:Thanks for your comments, we have changed the section and added more results in line 329-342, at the same time, we also included figure 5 to discuss more about the metabolites.
7. Line 394: Address safety concerns related to the colonization of E. faecalis.
A: Thanks for your comments, the strain of E. faecalis did not show harmful effects in mice, the results have been published in microoraganims 2024, here is the link https://www.mdpi.com/2076-2607/11/4/849.
8. Line 403: Compare findings with similar studies for better contextualization.
A:Thanks for your comments, we have added more information on the discussion part in line 468-488.
9. Table 1: Modify the title in "Differential Metabolites Identified in Fecal Samples of Mice Treated with Enterococcus faecalis DH9003."
A: Thanks for your comments, we have changed the table into a figure, which is figure 8 to demonstrate more about the results.
10. Figures 1 and 3, which show microbial diversity and phylum-level composition, could be presented together to allow direct comparisons between alpha diversity and relative abundances of key phyla. Include more detailed figure legends to explain the significance of each result, which would help readers interpret the findings more efficiently.
A: Thanks for your comment, Figure 1 is the alpha-diversity, but figure 3 is the composition shift of microbiota, since figure 2 represents the beta-diversity, it could be hard to combine those two figures. on the other hand, we include more detailed figures, such as figure5- figure8 to make the manuscript better.
Reviewer 3 Report
Comments and Suggestions for Authors
In the present article, the authors conducted further studies on the role of E. faecalis DH9003 in the regulation of gut flora in mice. Molecular biology tools were utilised to analyse the composition of the bacterial flora after oral administration of the bacteria. The work is conceptually interesting and well planned, with the illustration of the results being appropriate. Furthermore, the literature places the problem well in light of the findings of other researchers. Specific comments:
1. The chapter material and methods could be rendered more accessible by ensuring that the research is reproducible for other researchers, which is not currently the case due to the lack of a detailed description.
2. The authors use the term 'gut', which is quite broad and includes both the small intestine and the large intestine, and it would be beneficial to clarify this term to refer to the anatomically appropriate part of the large intestine.
Author Response
Dear reviewer,
The comments provided by you were very helpful to us. Based on your comments, we have carefully revised the manuscript (point by point). The revised parts are marked in the manuscript, and responses to the reviewers' comments are below.
1.The chapter material and methods could be rendered more accessible by ensuring that the research is reproducible for other researchers, which is not currently the case due to the lack of a detailed description.
A: Thanks for your comments, we have added more information in detailed in the Materials and methods section. All the modified parts are marked in yellow. hopefully, our additional information will satisfy your concerns.
2. The authors use the term 'gut', which is quite broad and includes both the small intestine and the large intestine, and it would be beneficial to clarify this term to refer to the anatomically appropriate part of the large intestine.
A:Thanks for your comments, we have added more information in line 369-375, trying to clarify that we focus on the "gut‘’ as in the large intestine.
Reviewer 4 Report
Comments and Suggestions for Authors
The aim of this study was to analyze reveals the roles of a bacteriocin-producing strain of Enterococcus faecalis DH9003 on regulatory effects of gut microbiota in mice. The results obtained are important for microbiologists. The research methods used are correct. Sufficient discussion. References selection correct and well used.
General comments:
I propose using the term "microbiota" throughout the article because "flora" - intestinal flora, gut flora are not accurate, bacteria are the animal world (fauna) and not the plant world (flora)
In the Introduction Section, please write the purpose of the research with the words "The aim of the research was..." and Research hypothesis - "The research hypothesis assumed......"
In the Materials and methods chapter there is no information about:
a. The method of housing mice, group cage, individual?, what size cages?
b. Please add information about the light program during the research period (length, intensity, color, type - incandescent, fluorescent?)
c. no information about the composition of the diet and information about water?
d. Were any bioactive preparations used during the study (probiotics, prebiotics, synbiotics, phytobiotics) that could have an effect on the microbiota beyond the experimental factor?
Others
For significance, use a low letter "p" in italic (p  0.05) instead of the "P" in the main article and space
The References section must be done according to the instructions for the authors
For all items, use a comma instead of a colon before the page range
Detailed comments:
L12 please add a phone number for the corresponding author
L70 Lac. – full name
L75 [18,19], no spaces
L102 15±1, no brackets
L182 small “p” in italic
L189 the groups? Or subgroups?
L206 Figure 1 instead of figure
L212, 249, 258, 262, etc. microbiota instead of flora
L2+665 space before (HD and YD)
L293 Lis. – full name
L349 Pan et al. [37]
L351 delete [37]

Author Response
Dear reviewer,
The comments provided by you were very helpful to us. Based on your comments, we have carefully revised the manuscript (point by point). The revised parts are marked in the manuscript, and responses to the reviewers' comments are below.
1.I propose using the term "microbiota" throughout the article because "flora" - intestinal flora, gut flora are not accurate, bacteria are the animal world (fauna) and not the plant world (flora)
A:Thanks for your comments, we have used the term “microbiota” throughout the whole manuscript.
2.In the Introduction Section, please write the purpose of the research with the words "The aim of the research was..." and Research hypothesis - "The research hypothesis assumed......"
A: Thanks for your comments, we have added the purpose of the research with the words "the aim of the study was ...." from line 92 to line 95.
3. In the Materials and methods chapter there is no information about:
a.The method of housing mice, group cage, individual?, what size cages?
A:we used M1 cage for housing the mice, the size of the cage was 290*180*160 mm. we grouped the mice into 2, one group was with 7 mice, and another group was with 8 mice. we have added the information in line 123.
b.Please add information about the light program during the research period (length, intensity, color, type - incandescent, fluorescent?)
A:Thanks for your comments, The mice were subjected to a 12-hour light/dark cycle (8 am to 8 pm exposed at a 15-watt incandescent light source), allowing them to eat and drink freely. the information was added in line 125-126.
c.no information about the composition of the diet and information about water?
A: thanks for your comments, we used SPF mouse feed. This feed is prepared particularly for C57 mice, under Co60 irradation. The composition of the feed was: corn, soybean meal, flour, bran, salt, multiple vitamins, amino acids, and multiple trace elements. The water we used was sterile water.
d.Were any bioactive preparations used during the study (probiotics, prebiotics, synbiotics, phytobiotics) that could have an effect on the microbiota beyond the experimental factor?
A:Thanks for your comments. We did not use other bioactive compounds during the whole study. On the contrary, we tried to minimize the extra interference from other compounds, including other bioactive substances. therefore, we conducted more procedures, like testing antimicrobial activity from fecal samples against Listeria monocytogenes, and continued to identify the strain by using PCR products. The results showed that only Enterococcus faecalis DH9003 was discovered in the study.
Others
For significance, use a low letter "p" in italic (p  0.05) instead of the "P" in the main article and space
The References section must be done according to the instructions for the authors
For all items, use a comma instead of a colon before the page range.
A:Thanks for your comments, we have changed all the error according to your comments.
Detailed comments:
L12 please add a phone number for the corresponding author.
A:Thanks for your suggestion, we have added a phone number for the corresponding author in line 13.
L70 Lac. – full name.
A: Thanks for your comments, we have changed it.
L75 [18,19], no spaces
A: Thanks for your comments, we have changed it.
L102 15±1, no brackets
A: Thanks for your comments, we have changed it.
L182 small “p” in italic
A: Thanks for your comments, we have changed it.
L189 the groups? Or subgroups?
A: Thanks for your comments, we have changed the groups into subgroups.
L206 Figure 1 instead of figure.
A: Thanks for your comments, we have changed it.
L212, 249, 258, 262, etc. microbiota instead of flora
A: Thanks for your comments, we have changed it.
L2+665 space before (HD and YD)
A: Thanks for your comments, we have changed it.
L293 Lis. – full name
A: Thanks for your comments, we have changed it.
L349 Pan et al. [37]
A: Thanks for your comments, we have changed it.
L351 delete [37]
A: Thanks for your comments, we have changed it.